# Association of Plant Protein Intake with Change in Physical Performance in Chinese Community-Dwelling Older Adults

**DOI:** 10.3390/nu14214534

**Published:** 2022-10-28

**Authors:** Suey S. Y. Yeung, Jean Woo

**Affiliations:** 1Department of Medicine and Therapeutics, Faculty of Medicine, The Chinese University of Hong Kong, Hong Kong, China; 2Jockey Club Institute of Ageing, The Chinese University of Hong Kong, Hong Kong, China; 3Centre for Nutritional Studies, Faculty of Medicine, The Chinese University of Hong Kong, Hong Kong, China

**Keywords:** physical performance, dietary protein, aged, Asian

## Abstract

(1) Background: Dietary protein intake might be beneficial in optimizing physical performance, yet whether this is dependent on protein source and sex is unclear. We examined the association between dietary protein intake and change in physical performance among Chinese community-dwelling older adults. (2) Methods: This prospective study included older Chinese adults (≥65 years) in Hong Kong. Total, plant and animal protein intakes at baseline were assessed using a food frequency questionnaire. Physical performance at baseline and 4-year follow-up were assessed by the time to complete a 6-m walking test. Adjusted linear regression examined the association between total, plant and animal protein intakes (g/kg of body weight (BW)) and 4-year change in physical performance. (3) Results: 3133 participants (49.8% males) aged 71.8 ± 4.9 years were included. In males, total, plant and animal protein intakes were not associated with a change in physical performance. In females, higher plant protein intake was associated with less decline in physical performance (β 0.723, SE 0.288, *p* = 0.012). No associations were observed for total animal protein intakes. (4) Conclusion: In Chinese community-dwelling older adults, total and animal protein intakes were not associated with a 4-year change in physical performance. Higher plant protein intake was associated with less decline in physical performance in females.

## 1. Introduction

The maintenance of physical performance with advancing age is a key component of healthy aging, which is defined as the process of developing and maintaining the functional ability that enables well-being in older age [1]. Physical performance measures have been suggested as a predictor of adverse outcomes such as dependence on activities of daily living, frailty, disability, cognitive impairment, poor quality of life, institutionalization, falls and/or mortality in community-dwelling older adults [2,3,4,5]. Understanding the lifestyle factors of decline in physical performance is essential to develop interventions that effectively preserve or improve physical performance. Apart from physical activity and medication use [6,7,8], diet has also been shown to be associated with physical performance in older adults.

Dietary protein intake has been the focus of nutritional epidemiology as amino acids (particularly branched-chain amino acids) from dietary protein are essential for muscle protein synthesis [9], thereby optimizing muscle function and physical performance [10]. A meta-analysis showed that a protein intake higher than the recommended dietary allowance of 0.8 g/kg body weight (BW) does not seem to prevent physical function decline over time in older adults [11]. The PROT-AGE study group recommends a dietary protein intake of 1.0 to 1.2 g/kg BW in healthy older adults [12]. Another meta-analysis showed that older adults with relatively very high (≥1.2 g/kg BW) and high (≥1.0 g/kg BW) protein intakes had better lower-limb physical functioning and mobility, respectively, compared to those with low protein intake (<0.8 g/kg BW) [13]. Most studies focused on meeting the daily protein requirement and used a specific cutoff based on sample distribution as the predictor [13,14,15,16,17,18,19,20]. This may not be necessarily useful when the protein intake of the studied population is relatively high, such as the Chinese older population (mean total protein intake of 1.2 to 1.6 g/kg BW) [21,22,23]. Furthermore, animal protein is generally considered a more potent stimulator of muscle protein synthesis than plant protein. However, several cross-sectional studies from Western countries showed that higher plant but not animal protein intake was associated with better physical performance [14,24,25].

In view of the scarcity of evidence on this topic in Asia, this study examined the association between dietary protein intake (total, plant and animal) and change in physical performance among Chinese community-dwelling older adults from the Mr. and Ms. Osteoporosis cohort (Os) in Hong Kong.

## 2. Materials and Methods

### 2.1. Study Design

A total of 4000 Chinese community-dwelling older adults aged 65 years and above were recruited in Hong Kong between 2001 and 2003 [26]. Participants were volunteers who were able to walk or take public transport to the study site to attend a health check. Exclusion criteria included a bilateral hip replacement or inability to give informed consent. A stratified sample was applied to recruit approximately 33% of the participants for each of the age groups: 65–69, 70–74, 75+. Participants were followed up by a visit to the study site 4-year after baseline. Participants were excluded from the analysis due to missing data on protein intake (*n* = 5), extreme energy intake (i.e., <500 kcal or >5000 kcal) (*n* = 3), and incomplete measurement of the 6-m walking test at baseline and 4-year follow up (*n* = 859). The final sample size for the present analysis was 3133 (1560 males and 1573 females). This study was conducted in accordance with the Declaration of Helsinki. The Clinical Research Ethics Committee of a local institution has granted approval for this study (CREC 2003.102). All participants completed the written informed consent.

### 2.2. Characteristics of Participants

A standardized interview was conducted to collect information on age, sex, body weight, body mass index (BMI), education level (primary or below/secondary/university or above), cigarette smoking status (current/no or past), alcohol use (drink >12 alcoholic drinks in past 12 months: yes/no), number of diseases and physical activity level. Body weight in kilograms was measured using the Physician Balance Bean Scale (Healthometer, Bridgeview, IL). Participants were asked to wear light clothing. Height was measured using the Holtain Harpenden Standiometer (Holtain Ltd., Crosswell, UK). BMI was calculated as body weight (kg) divided by height in meters squared (m^2^). The number of diseases was assessed with the use of a pre-defined list, in which participants were asked if they had ever been told by a healthcare professional that they had or have any of the diseases on the list. Supplementary information on medical history was provided by the identification of medications brought to the interviewers. The Physical Activity Scale for the Elderly (PASE) was used to assess physical activity levels [27]. The PASE includes 12 items assessing the average number of hours per day spent in leisure, household, and occupational physical activities over the past seven days. Each item score was computed by multiplying the activity weight (determined with reference to the amount of energy spent) with daily activity frequency reported. A PASE score was calculated by summing each item score. A higher PASE score represents a higher physical activity level.

### 2.3. Dietary Assessment

Dietary intake was collected using a validated 280-item food frequency questionnaire (FFQ), which was originated and validated in a population-based survey among local participants aged 25 to 74 years [28]. Participants were instructed to complete the FFQ to provide information on consumption frequency and the quantity of each listed food item, using the past 12 months as a reference period. A catalog of pictures was used to explain the individual food portions. To estimate the quantity of cooking oil used, the usual methods in preparing standardized portions and the usual portions of different foods consumed by the participants were taken into consideration. Food composition tables compiled from McCance and Widdowson [29] and the Chinese Medical Sciences Institute [30] were used to obtain nutritional information and calculate the mean daily energy (kcal) and protein intake (g). Protein intake (g) is additionally expressed as protein intake relative to body weight (g/kg BW). Total protein intake included plant and animal proteins. Percentages of protein intake from plant/animal sources were calculated by the quantity of plant/animal protein (g) divided by the total protein intake (g). Plant proteins refer to proteins obtained from plant sources such as cereals, fruits and dried fruits, legumes, seeds and nuts, and vegetables. Animal proteins refer to proteins originating from animal sources such as meat and poultry, fish and shellfish, eggs and milk and milk products. Since nutrients in foods are highly correlated and nutrients other than protein may contribute to the association, we considered the overall diet quality of the participants. Diet quality was examined using the Diet Quality Index-International (DQI-I), which has been applied as a dietary quality measure in a Chinese population and is an indicator of dietary patterns in relation to health [31,32]. The DQI-I includes components such as fatty acid ratio, adequacy of vitamin C intake and variety within the protein group.

### 2.4. Physical Performance

Physical performance was assessed using a 6-m walking test at baseline and a 4-year follow-up. Participants were instructed to walk along a straight line 6-m long in distance at their usual pace. Timing was started when the participant’s foot touched the floor on the first step. Timing was stopped when the participant’s foot touched the floor at the finish line. Two trials were performed, and the best time in seconds to complete the walking test was used for analysis. Shorter time to complete the walking test reflected better physical performance.

### 2.5. Statistical Analysis

Data are presented as means and standard deviations (SD) for continuous variables and frequency and percentage for categorical variables where appropriate. Independent t-tests were used to examine the differences in protein intake and change in physical performance by sex.

Change in physical performance was computed as baseline value minus 4-year follow-up value (a negative value indicates more time taken to complete the test and thereby a decline in physical performance). Multivariable linear regression was used to examine the association between protein intake and change in physical performance. To control for the possible effect of individual differences in protein quantity on the associations, we used the residual method to adjust for the daily total protein intake for the analysis of plant protein and animal protein [33]. In brief, to calculate protein-adjusted intake of plant/animal protein, the residuals for each individual from the linear regression model with plant/animal protein intake (g/kg BW) as the dependent variable and total daily protein intake as the independent variable were added to a constant, which is the expected plant/animal protein intake for the mean total protein intake of the study population. Analysis was adjusted for age (continuous), the number of chronic diseases (continuous), daily energy intake (continuous), DQI-I score (continuous), current smoker (yes/no), current drinker (yes/no), PASE score (continuous) and amount of medications (continuous). These factors were chosen as the literature suggested that they were considered potential risk factors for physical performance [6,7,8,34,35,36]. Results are presented as regression coefficients (β) and standard error (SE). Sex differences in the association between diet and physical performance have been reported previously [17,37,38,39,40]; therefore, a priori sex-stratified analyses were conducted. All analyses were conducted using the statistical package SPSS version 26.0 (IBM SPSS Statistics for Windows, Version 26.0. Armonk, NY, USA: IBM Corp.). A two-sided *p*-value of <0.05 was considered statistically significant.

## 3. Results

Table 1 shows the characteristics of the participants. A total of 3133 participants (49.8% males) aged 71.8 ± 4.9 years were included in the analysis. The mean BMI was 23.7 kg/m^2^. A majority of the participants had an education level of primary or below (70.3%), were non-current smokers (93.8%) and non-current drinkers (86.5%). The mean total, plant and animal protein intakes were 1.3 ± 0.6, 0.6 ± 0.3 and 0.7 ± 0.4 g/kg BW, respectively. An average of 46 ± 14% of the total protein intake was derived from plant sources, with a significant sex difference (48.7 ± 13.8% for females vs. 44.0 ± 13.6% for males, *p* < 0.001). Rice (14 ± 8% of total protein intake) and soy and soy products (6 ± 5% of total protein intake) were the main sources of plant protein, while meat and poultry (22 ± 11% of total protein intake) and fish and shellfish (19 ± 11% of total protein intake) were the main sources of animal protein. The mean (SD) 4-year change in time to complete the 6-m walking test was an increase of 0.9 ± 2.2 s, with females showing a significantly greater change than males (increase of 1.0 ± 0.3 vs. 0.7 ± 2.0 s over 4-year).

Table 2 shows the associations between baseline relative protein intake and change in physical performance. In males, higher baseline total, plant and animal protein intakes were not associated with a change in physical performance in unadjusted and adjusted models (all *p* > 0.05). In females, higher baseline plant protein intake was associated with less decline in physical performance (adjusted β 0.723, SE 0.288, *p* = 0.012) over 4 years. No associations were found between total protein and animal protein intakes and changes in physical performance.

## 4. Discussion

In Chinese community-dwelling older adults, total protein intake was not associated with a 4-year change in physical performance. The association between protein intake and change in physical performance was dependent on the protein source and sex. Specifically, animal protein intake was not associated with a 4 year change in physical performance in both sexes, while higher plant protein intake was associated with less decline in physical performance in females only.

To the best of our knowledge, studies examining the prospective association between total, plant and animal protein intakes and physical performance were limited. Prospective analysis using specific cutoffs of protein intake as predictors showed that adequate total protein intake (≥1.0–1.2 g/kg BW) was not associated with a 3-year change in the 10-m walking test performance [16] and 5-year change in Timed Up-and-Go test performance in older adults [17]. In contrast, a pooled analysis of four longitudinal aging cohorts showed that higher daily protein intake (≥0.8 g/kg BW) was protective of the decline in walking speed over a follow-up of a maximum of 8.5 years [20]. Several cross-sectional studies supported our findings. Relative total protein intake was not associated with gait speed among well-functioning older adults [25,41]. It has been suggested that the lack of association might be caused by a ceiling effect in the chosen test battery, as the mean gait speed of 1.0 m/s was high compared to other studies [41]. Regarding the association between protein source and physical performance, higher plant protein intake was associated with higher gait speed at a fast pace in well-functioning older adults [25]. In contrast, animal and plant protein intakes were not associated with the walking test among Dutch older adults [42]. Lower animal but not plant protein intake was associated with a greater risk of mobility limitation over a 6-year of follow-up in the Health ABC study [43].

The proportion of plant protein intake in our study was relatively high compared with Western countries (ranging from 24% to 39% plant proteins) [14,44,45,46,47,48]. The main sources of plant protein were rice and soy and soy products in our Chinese population, while cereal is the principal plant protein source in Western countries [45,48]. This study found that plant protein but not animal protein intake was associated with a change in physical performance. Animal protein contains high leucine content and a complete amino acid profile required for muscle protein synthesis, which has been considered more anabolic than plant protein [49]. However, the notion that animal protein is key for muscle health has been challenged [24,25,42]. Plant-based diets have an alkaline property and are rich in potassium and magnesium, which favors the maintenance of muscle mass and function in older adults [50,51]. In contrast, an animal-based diet increases the production of acid in the body, and low-grade metabolic acidosis may be detrimental to muscle mass and function [52]. It has been suggested that diets high in plant protein sources have the potential to support the maintenance of muscle mass with aging when sufficient amounts of protein are consumed [49,53]. Consuming greater amounts of protein may compensate for the lower essential amino acid content of plant protein [54]. Furthermore, in a plant-based diet, incomplete protein can be combined to form complete essential amino acid profiles that resemble animal protein [54,55]. Of note, the quality of plant protein depends on the food source. For example, legumes such as soybeans and peas are high in leucine, which has been demonstrated to maximize muscle protein synthesis [56,57]. It has been suggested that protein and amino acid intakes were adequate when plant protein is up to 50% of total protein. When the relative intake of plant protein is lower than 70% of total protein intake, no deficits regarding essential amino acids would be expected to occur [48]. Plant protein contributed 46.4% of the total protein intake in our population; therefore, there should be no negative impact on the adequacy of protein and essential amino acid intakes. Furthermore, total protein intake (1.3 ± 0.6 g/kg BW) in our population was relatively high compared with other older populations [16,19,25,38,39,58,59,60], and 69% of our participants met the recommended protein intake of at least 1.0–1.2 g/kg BW [12]. Together with adequate total protein intake, the beneficial role of a high proportion of protein from plant sources on muscle health may be enhanced and therefore explain the association of plant protein with less decline in physical performance in this study.

Due to the observational nature of our study, the causal relationship between plant protein intake and a decline in physical performance cannot be drawn. A systematic review and meta-analysis summarized the intervention studies, which compared the potential differences in the effect of animal vs. plant protein on muscle-related outcomes [61]. It was shown that a significant effect favoring animal protein intake was observed in young adults for lean mass but no such effect in muscle strength outcomes. However, animal protein intake did not show a favorable effect on muscle-related outcomes in older adults [61]. Although more evidence from intervention studies is required, a plant-based diet appears to adequately preserve muscle-related outcomes in older adults, provided that the total protein intake is high. Soy and soy products, such as tofu, dried tofu sheets and soy milk, are inexpensive and readily available from the market, making them a good alternative for animal protein among older Chinese adults.

The sex difference in the association between protein intake and physical performance was partly consistent with previous studies. A similar study among older adults in the UK showed an association between nutrient intake and physical performance (3-m walk) in women but not in men [37]. The Framingham Heart Study Offspring cohort showed that higher protein intake was associated with a 30% lower risk of loss of functional integrity (17 measures of function) in women but not in men [19]. Among diabetic older adults from the NuAge cohort, adequate total protein intake (≥1.0 g/kg BW) was associated with the maintenance of functional capacity in women but not in men [40]. As women have poorer physical performance compared with men, it is proposed that there may be a threshold below which variations in diet are of greater importance [37]. This was also reflected in our cohort, in which females have a significantly greater decline in walking test performance compared with males. The higher regression coefficient in females compared with males reported in this study reflect that protein intake had a stronger association with physical performance in older females compared with males. Furthermore, there is evidence to suggest that antioxidant intakes may affect physical performance in older adults [62]. As females consumed a larger proportion of plant protein compared with males in our study, the antioxidant intake from plant-based foods may have an additional effect on physical performance in addition to plant protein per se.

Our study has several strengths and limitations. Objective physical performance measures were used. The use of FFQ also allows the assessment of usual protein intake and a range of protein sources. Several potential confounding factors have been collected and included in the analysis. However, the current study included exclusively Chinese; therefore, it limits the generalizability to other ethnic groups. Second, we did not account for the potential changes in diet over time. Third, self-reported FFQ is subject to recall bias and does not provide information on mealtime distribution of protein intake. Our study population included volunteers who were willing to visit the study site and, therefore, may be more health-conscious and have a better diet than the general population. Finally, a causal relationship cannot be established as this was an observational study. Further trials investigating the protective role of plant protein intake in muscle health are warranted. The underlying mechanism responsible for the beneficial effects of plant protein in comparison to animal protein also needs to be explored in future studies.

## 5. Conclusions

In Chinese community-dwelling older adults, total protein intake was not associated with a 4-year change in physical performance. The association between dietary protein intake and change in physical performance was dependent on protein source and sex. Animal protein intake was not associated with a 4-year change in physical performance in both sexes. Higher plant protein intake was associated with less decline in physical performance over 4-year in females only. This study provides insight into how different sources of protein intake might influence physical performance in older adults. It appears that plant protein intake can be encouraged for the optimization of physical performance among older Chinese adults with a dietary culture of high total protein intake.

## Figures and Tables

**Table 1 nutrients-14-04534-t001:** Baseline characteristics and 4-year change in physical performance of participants.

	All (*n* = 3133)	Male (*n* = 1560)	Female (*n* = 1573)
Age, years	71.8 ± 4.9	71.8 ± 4.7	72.0 ± 5.0
Body weight, kg	58.7 ± 9.6	62.7 ± 9.2	54.8 ± 8.3
BMI, kg/m^2^	23.7 ± 3.2	23.5 ± 3.1	24.0 ± 3.4
Education level, %			
Primary or below	70.3	58.6	81.9
Secondary	19.2	26.8	11.8
University or above	10.5	14.6	6.4
Current smoker, %	6.2	10.8	1.7
Current drinker, %	13.5	24.4	2.7
Number of chronic diseases	1.9 ± 1.5	1.9 ± 1.5	1.9 ± 1.4
PASE, score	94 ± 43	101 ± 51	87 ± 33
DQI-I (0–94), score	65 ± 9	64 ± 9	66 ± 9
Energy intake, kcal/d	1856 ± 585	2115 ± 584	1598 ± 458
Absolute protein intake, g/d			
Total	78 ± 33	89 ± 35	66 ± 28
Plant	35 ± 17	38 ± 18	31 ± 14
Animal	43 ± 25	51 ± 26	35 ± 21
Relative protein intake, g/kg of BW/d			
Total	1.3 ± 0.6	1.4 ± 0.6	1.2 ± 0.6
Plant	0.6 ± 0.3	0.6 ± 0.3	0.6 ± 0.3
Animal	0.7 ± 0.4	0.8 ± 0.4	0.6 ± 0.4
% Protein from plant sources	46.4 ± 13.9	44.0 ± 13.6	48.7 ± 13.8
Gait speed, m/s	1.03 ± 0.22	1.09 ± 0.22	0.98 ± 0.20
Change in time to complete 6-m walking test, seconds	−0.9 ± 2.2	−0.7 ± 2.0	−1.0 ± 2.3

BMI, body mass index; PASE, Physical Activity Scale for the Elderly; DQI-I, Dietary Quality Index-International; BW, body weight.

**Table 2 nutrients-14-04534-t002:** Associations between baseline relative protein intake (g/kg of body weight) and change in physical performance over 4 years.

	4-Year Change in Time to Complete 6-m Walking Test ^1^					
	Unadjusted			Adjusted ^2^		
	β	SE	*p*	β	SE	*p*
Total protein						
Males (*n* = 1560)	0.120	0.088	0.170	0.038	0.130	0.767
Females (*n* = 1573)	−0.118	0.103	0.256	−0.158	0.154	0.306
Plant protein ^3^						
Males (*n* = 1560)	0.332	0.216	0.125	0.301	0.232	0.195
Females (*n* = 1573)	0.496	0.273	0.070	0.723	0.288	0.012
Animal protein ^3^						
Males (*n* = 1560)	0.063	0.119	0.598	0.045	0.209	0.830
Females (*n* = 1573)	−0.131	0.200	0.512	−0.305	0.251	0.225

^1^ Baseline value minus 4-year value; ^2^ adjusted for age, number of chronic diseases, daily energy intake, DQI-I, PASE score, current smoker, current drinker and number of medications; ^3^ plant/animal protein were adjusted for total protein intake using the residual method.

## Data Availability

The data presented in this study are available on request from the corresponding author.

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
