# Peer review of "Association of Plant Protein Intake with Change in Physical Performance in Chinese Community-Dwelling Older Adults"

_nutrients, 2022, doi:10.3390/nu14214534_

Round 1

Reviewer 1 Report

The manuscript titled "Dietary intake of plant but not animal protein is associated with physical performance change in Chinese community-dwelling older adults" describes the association between animal vs plant protein and physical performance changes over 4 years in males and females.  

The unadjusted vs adjusted demonstrated completely different findings for females.  This needs to be explained - specifically adding justifications for each adjusted variable.  The results section needs better explanations.

The intro/methods/discussion are sufficient.  This paper just feels weak due to the results section.

Reviewer 2 Report

The authors followed over 3000 study subjects for 4 years. Observing a large number of research subjects over a long period of time is a very precious thing. However, the observations were superficial. It is difficult to find new facts in this study about how the differences in dietary ingredients are established.

Reviewer 3 Report

The paper is very interesting and it discuss a very important topic concerning the relationship between nutrition a physical performance in older adults.
The topic is particularly interesting expecially for its impact on healthcare system and quality of life of elderly patients. I suggest to change the title, the present one is confusing and not incisive, i suggest: " Association of dietary intake of plant protein with physical performance in Chinese community dwelling older adults" Moreover I suggest: - to consider the impact of physical activity on physical performance (10.1016/j.archger.2020.104109 and 10.1016/j.jamda.2019.01.128)
- consider the impact of polymedication on older adults and their performance (10.1017/S1041610217001715 and 10.1007/s40520-018-0893-1) Thanks    

Round 2

Reviewer 1 Report

These revisions to support the results analysis are sufficient.

Reviewer 2 Report

The authors have modified the title, introducton and methods but not the results and discussions. However, there is no change in the fact that there are no new facts in the data analysis and that the discussion is insufficient.